# A blank check or a global public good? A qualitative study of how ethics review committee members in Colombia weigh the risks and benefits of broad consent for data and sample sharing during a pandemic

**María Consuelo Miranda Montoya**[1], **Jackeline Bravo Chamorro**[1], **Luz Marina Leegstra**[2], **Deyanira Duque Ortiz**[3], **Lauren Maxwell**[2]*

1 Facultad de Salud, Universidad Industrial de Santander, Bucaramanga, Santander, 2 Heidelberger Institut für Global Health, Universitätsklinikum Heidelberg, Heidelberg, Germany, 3 Ministerio de Ciencia, Tecnología e Innovación MINCIENCIAS, Bogotá D.C., Colombia

* lauren.maxwell@uni-heidelberg.de

## Abstract

Broad consent for future use facilitates the reuse of participant-level data and samples, which can conserve limited resources by confirming research findings and facilitate the development and evaluation of public health and clinical advances. Ethics review committees (ERCs) have to balance different stakeholder concerns when evaluating the risks and benefits associated with broad consent for future use. In this qualitative study, we evaluated ERC members' concerns about different aspects of broad consent, including appropriate governance, community engagement, evaluation of risks and benefits, and communication of broad consent for future use in Colombia, which does not currently have national guidance related to broad consent for future use. We conducted semi-structured, in-depth interviews with 24 ERC members from nine Colombian ERCs. We used thematic analysis to explore ERC members' concerns related to broad consent for future use. Most ERC members expressed concern about the idea of not specifying the purposes for which data would be used and by whom and suggested that pre-specifying governance procedures and structure would address some of their concerns about broad consent. ERC members emphasized the need for engaging communities and ensuring research participants understood broad consent for future use-related language in informed consent forms. Researchers and research institutions are under increasing pressure to share public health-related data. ERC members play a central role in balancing the priorities of different stakeholders and maintaining their community's trust in public health research. Further work is needed on guidelines for developing language around broad consent, evaluating community preferences related to data sharing, and developing standards for describing governance for data or sample sharing in the research protocol to address ERC members' concerns around broad consent for future use.

**Data Availability Statement:** The study protocol, demographic information sheet, and IDI guide are available on the Open Science Foundation (10. 17605/OSF.IO/2YUVF). Relevant, deidentified, excerpts in the original Spanish and the corresponding English language translation are included in the manuscript and Supporting information files. Due to the descriptive nature of the data, the sensitive nature of the topic for ERC members, who may receive pressure from researchers and policy makers to enable data sharing, and the small sample size of the source population (ERC members in Colombia), the complete Spanish-language deidentified interview transcripts will not be uploaded to a data sharing registry to protect participant privacy.

**Funding:** This study received funding support from TDR, the Special Programme for Research and Training in Tropical Diseases, grant to LM [P20-00007] and the ReCoDID project, funded by the EU Horizon 2020 research and innovation programme (grant agreement 825746) and the CIHR Institute of Genetics (grant agreement 01886-000) grant to LM. The funders of the study had no role in study design, data collection, data analysis, data interpretation, or writing of the report.

**Competing interests:** The authors have declared that no competing interests exist.

# Introduction

The global public health community has long recognized the need to share participant-level data and biological samples to facilitate the development of diagnostics, prophylaxis, and treatments during and outside of epidemic response [1,2]. Data sharing can facilitate reproducibility and trust in the research process and in research outputs; conserve limited resources through the prevention of redundancies, and build cross-national and cross-discipline networks which can accelerate discovery and innovation. In the best case, data sharing can save lives through more efficient and effective research public health response [3]. Conversely, data sharing can lead to parachute research where researchers who are unfamiliar with the context and data can misinterpret the data; and to inequities where data collected from one population benefit another population or data are exploited for commercial purposes with no benefit to the source population. Data sharing without clear and ethical governance can engender mistrust between researchers and the communities that participate in research or widen inequities as when samples from one population are used to develop treatments or vaccines that benefit another population.

Before initiating the exchange of data and samples or the construction of a biobank that serves research groups at different institutions within or across countries, different ethical, legal and social aspects of data and sample sharing must be considered [4,5]. International collaborations between high- and low-and-middle-income countries (LMICs) support research in understudied populations and facilitate the exchange of participant-level data and biological samples.

Ethics review committee (ERCs) and institutional review board (IRBs) members play a central role in balancing the priorities of different stakeholders and maintaining their community's trust in public health research. Review of study protocols by ERCs and IRBs strengthens research protocols through clarifying the risks and benefits of the proposed research and ensuring best practice in data and sample sharing, including clear governance for management of sensitive data and samples, engagement of source communities, and benefit sharing. Conversely, ERC-related delays in the approval process can unduly burden research teams and slow much needed research, particularly during an epidemic [6].

## Broad consent for future use

Broad consent for future use is research participants' consent for a loosely or unspecified range of future research purposes that go beyond the objectives of the original study [7]. Participants who provide broad consent are not re-contacted when their data or samples are used in additional studies. The inclusion of broad consent in the informed consent form (ICF) facilitates future collaborations and the reuse of valuable participant-level data. Broad consent for future use must meet the Council for International Organizations of Medical Sciences' (CIOMS) criteria for the application of broad consent, including appropriate governance and that sharing data or samples must not adversely affect the rights and welfare of research participants [8]. Where broad consent was not obtained, CIOMS presents guidance for ERC members' consideration of an informed opt out procedure or a waiver of consent [8].

While researchers are under increasing pressure from funders and journals to ensure that de-identified, participant-level data from their studies can be shared for future analyses, ERC members may not allow the inclusion of broad consent for future use when research studies do not appropriately address the CIOMS criteria or when they have outstanding concerns about benefit sharing, equity, and participant privacy. ERC members may be uncomfortable with the permissive language related to broad consent, especially in the absence of national or local guidelines. Because clinical and cohort studies provide valuable longitudinal data on a

number of factors beyond the exposure of interest for the initial study, studies that do not include broad consent for future use in their ICFs represent a missed opportunity to maximize the use of data and samples and may lead researchers to collect new data to answer questions that could otherwise be answered by existing data.

## The legal and policy context for broad consent in Latin America and the Caribbean

A 2018 Pan American Health Organization (PAHO)-led meeting of Latin American and the Caribbean (LAC) countries identified developing a systematic approach to research ethics, that included capacity building and guidelines for ERC review during public health emergencies, as a regional priority, but neither broad consent nor data or sample sharing were mentioned in the related meeting resolution [9]. A recent review of national guidance from 19 of the 33 countries in LAC related to policies for expediting review of COVID-19-related human subjects research during the pandemic found that Colombia was among the 10 LAC countries that implemented such guidance [10]. While future use was among the topics considered in six LAC countries that issued COVID-19-related guidance to ERCs, Colombia was not one of these countries [10]. These findings reflect a similar policy analysis conducted in 22 African countries which found that only three countries had national or local guidelines related to biobanking and genetic research [11]. Three African countries had laws or guidelines related to broad consent for future use, which was not allowed or required reconsenting subjects in those countries [11].

## The legal context for broad consent in Colombia

Colombia has a national legal framework for evaluating ethical concerns related to the interface of research and medical practice but does not have a national legal framework to guide the consideration of protocols that include broad consent for future use. That said, the Colombian legislature has been working to regulate the establishment of biobanks and the use of samples and biological collections in research [12]. The bill "By means of which the operation of Biobanks for biomedical research purposes is regulated and other provisions are enacted" ("Por medio del cual se regula el funcionamiento de los Biobancos con fines de investigación biomédica y se dictan otras disposiciones"), which addresses the ownership and sharing of samples, was filed by the legislature in 2017 [12] but has not been considered further. Subsequently, legislative proposal 168, to"regulate the constitution and operation of biobanks for obtaining, processing, storage, transport and transfer of human biological samples, their derivatives, associated clinical and biological information for biomedical research purposes" was submitted for review by the Senate of Colombia in 2019. This qualitative study helps to elucidate whether, in the absence of national guidance in Colombia, ERC members' concerns related to the application of CIOMS guidance on broad consent for future use (e.g. governance, protection of research participants anonymity and welfare) or the lack of national guidelines related to the management of data and sample sharing.

## Materials and methods

We conducted in-depth interviews (IDIs) using the Zoom video conference platform with 24 members of 9 ERCs from five large cities in Colombia (Bogotá, Cali, Barranquilla, Medellín, Bucaramanga) to 1) understand ERC members' primary concerns when evaluating research protocols that include broad consent for future use or applications for waivers of consent, and how these concerns vary by the type of data or samples being shared, with whom the data may be shared, and from whom the data are collected; 2) explore how ERC members balance the

competing priorities of different stakeholder groups when making decisions related to broad consent for future use and waivers of consent; and 3) understand whether ERC members of different genders, professional backgrounds, career stages, or roles on the ERC express different concerns around broad consent for future use and waivers of consent. The semi-structured interview guide was developed by a group of social scientists, ERC members, and legal scholars whose research relates to data and sample reuse in Colombia and internationally. The research team reviewed related literature and consulted with researchers who work on data and sample sharing in several different LMICs and globally. The external review resulted in (1) the rewording of the study objective in the participant information sheet to take a more neutral approach towards broad consent for future use and data and sample sharing more generally and (2) the reorganization of the interview guide, including the removal of redundant questions and a reduction in the number of probes. The interview guide was piloted with three participants. Interviews were conducted in Spanish and recordings were transcribed verbatim.

### Identification & recruitment of study participants

Colombia does not have a public database of existing ERCs and their membership. The study team identified ERCs that review interventional or observational research that includes broad consent for future use from the Instituto Nacional de Vigilancia de Medicamentos y Alimentos (INVIMA) institutional page which lists Good Clinical Practice-certified ERCs. Invitation letters were sent by email to the ERC Chairs of 20 public and private institutions in the Central, Southwest, Northeast, Andean and Caribbean regions. We then contacted the ERC president for the 12 ERCs that responded to the initial request. In some cases, the president requested a formal presentation of the project by the Colombia-based project team, which includes an epidemiologist and physician-researcher who has served as a member of ERCs and a philosopher who focuses on bioethics from the Ministry of Science, Technology and Innovation. The first contact or formal presentation was followed by an internal discussion by ERC members related to whether the ERC would participate and, if so, which members would participate in the study. In some cases, the ERC's nominated participants who were more experienced with reviewing protocols that include broad consent for future use. At least two members of each of the 9 ERCs that agreed to participate were interviewed for the study. We considered two sources of heterogeneity when recruiting the sample: 1) role on the ERC (Leadership (e.g. Chairperson, Secretary), Scientific member, Non-scientific/community members) and 2) gender.

### Data collection

Interviews were conducted between 15 January and 10 March 2021. The one-time interview lasted for about one hour. Interviews were conducted until saturation was reached, i.e., no new themes emerged.

### Researcher characteristics and reflexivity

Participants were recruited by an infectious disease physician (MCMM) who has served on several ERCs in Colombia and by a philosopher who is a member of the Ministry of Science and Technology and has been working with government officials on legislation related to data and sample sharing (DDO). The member of the Ministry of Science and Technology and a nurse who has a Master's degree in public health ethics (JBC) conducted the phone-based interviews. We realized during the team debriefs of the first several interviews that the inclusion of a member of the Ministry of Science and Technology as an interviewer made some participants feel like their knowledge of broad consent for future use was being evaluated. The member of the Ministry of Science and Technology did not conduct further interviews and the

interviews that had been conducted were compared between the two interviewers for differences in themes. We did not find differences in the way ERC members evaluated broad consent for future use or their concerns between the two interviewers. Although the formal presentation of the research project to ERCs was deemed a necessary professional curtesy, this may have led potential participants to believe that their knowledge of broad consent would be evaluated. We tried to address this concern by stating that there were no right or wrong answers in the presentation to the ERCs and the informed consent for the interview. All research team members are engaged in projects related to data and sample sharing, with a focus on data sharing in epidemic response. As such, we asked colleagues from the infectious disease field to review the guide to ensure that the interview guide language and questions were not biased in favor of data and sample sharing.

## Ethics statement

Acting ERC members provided their verbal consent to participate in the study prior to beginning the interview. Participants did not receive any incentive for participation. Participant names and any other identifying information were removed from the written transcripts. The research protocol was approved by the Ethics Review Committee of the Centro de Atención de Diagnóstico de Enfermedades Infecciosas, Bucaramanga, Colombia. Further ethics considerations are reported in the PLOS Inclusivity in global research questionnaire (S1 Text).

## Data analysis

The written, de-identified transcripts were uploaded to MAXQDA [13] software and analyzed using a combined deductive and inductive approach rooted in Colaizzi's approach to descriptive phenomenology [14,15]. During the first phase of the analysis, we developed an initial set of deductive codes and definitions based on (1) the interview guide; (2) literature identified through a concurrent scoping review related to broad consent for future use [16] and (3) our own experience in research ethics as members of ERCs and as researchers in the field of public health ethics and data sharing. Inductive codes were identified through a review of the written transcripts and interviewer field notes. Transcripts were read several times so that the team could familiarize themselves with the data and identify important statements which were then interpreted by the research team. All authors reviewed three transcripts together to refine the codebook and definitions. To ensure consistency in the interpretation of codes, two researchers (JBC, LML) coded several additional interview transcripts together. To ensure reliability, codes were independently applied to blocks of text in the interview transcripts by two researchers (JBC, LML) who met weekly to discuss and resolve differences in the application of codes and discuss emergent inductive codes, grounded in the transcript data. During the same period, the entire team held weekly discussions to update and refine the codebook and to ensure agreement on the application of the codes across all team members. This process meant that two team members independently reviewed all transcripts and a subset of transcripts was reviewed and evaluated by all four members of the team. Team members documented the evolution of codes through memos and meeting notes and compared the application of codes in later transcripts to that of earlier transcripts to ensure consistent application of codes (reliability) throughout the first phase of the analysis. Earlier interviews were recoded to account for emerging codes and evolving definitions.

During the second phase of the analysis, the team met weekly to group the codes into meaningful themes and to discuss whether codes and related themes varied across participants of different genders, levels of experience on ERCs, and positions within the ERC. Reliability was

ensured through weekly meetings to discuss the evolution and application of codes and their definitions between the coders and with the entire team. The interview guide was modified to explore emerging themes, including the tension between an individual's beliefs and values and the group's beliefs and values. Validity was ensured through grounding deductive codes in the findings from a concurrent scoping review and team members' own experience as ERC members and researchers in the field of bioethics and data and sample sharing and in the use of the interview transcripts to identify emerging (inductive) codes. Throughout the project, differences of opinion in the development and application of codes and themes were resolved by consensus. Results are presented in keeping with the Standards for Reporting Qualitative Research [17]. The IDI guide is included as S2 Text.

## Results

Participant demographics and related training and experience are presented in Table 1. More than half of the participants were older than 45 (58%); 38% were between 35–45 years, and 4% were under 25 years old. Most participants (63%) were female and half of all the participants were members of an ERC at an academic institution (50%). Outside of their service to the ERC, participants worked as medical professionals (58%), social scientists or lawyers (13%), and in the life sciences (13%). Most participants (70%) had at least five years of experience on an ERC and most had reviewed a protocol that included broad consent for future use (80%). Most participants (67%) reported their ERC did not have a procedure to review research protocols in the context of a pandemic or epidemic. Half (N = 12) of participants said that the international guidelines followed by their committee allow broad consent for future use, while 38% said not and 12% did not answer that question. Most of the ERC members reported receiving ethics-related training (84%) through postgraduate studies, clinical good practices, and seminars and training from their institutions and other national and international institutions. Participants described a number of risks and benefits related to broad consent for future use for the participant, research team, and community described below.

### Lack of national guidance on broad consent for future use

ERC members agreed that, at the national level, the lack of laws and policies regarding sharing data and samples from human subjects' research for use in future studies complicates the review process.

*"We don't have anything to hold on to normatively, there is nothing. Let's say it is important to discuss this at some point"*

*(CEI-020, male, President, 35–45 years old)*

In lieu of national guidance, ERC members reported using international guidelines to inform protocol review.

*"Let's say here how there is no biobanking law or a specific norm. . .there are some indirect references that are not referents for research but help."*

*(CEI-020, male, President, 35–45 years old)*

Beyond the lack of national guidance, participants reported that sometimes not having the necessary expertise limited participation in the review of some protocols.

**Table 1. Participant characteristics (N = 24).**

| | N | % |
|---|---|---|
| Location | | |
| Cali | 4 | 17% |
| Bogotá | 10 | 41% |
| Bucaramanga | 5 | 21% |
| Barranquilla | 3 | 12% |
| Medellín | 2 | 9% |
| Age | | |
| under 25 | 0 | |
| 25–34 | 1 | 4% |
| 35–45 | 9 | 38% |
| over 45 | 14 | 58% |
| Gender | | |
| Male | 9 | 37% |
| Female | 15 | 63% |
| Other | 0 | |
| Professional background | | |
| Health professional | 14 | 58% |
| Life sciences | 3 | 13% |
| Social sciences | 3 | 13% |
| Economical sciences | 2 | 8% |
| Non-biomedical sciences | 0 | |
| Other | 2 | 8% |
| ERC institutional affiliation | | |
| Academic | 12 | 50% |
| Health care institution | 2 | 8% |
| Public health institution | 3 | 13% |
| Public and health care institution | 7 | 29% |
| Position in the ERC | | |
| Chairperson | 5 | 22% |
| Secretary | 3 | 13% |
| Other members | 11 | 48% |
| Community representative | 4 | 17% |
| Years of experience as an ERC member | | |
| Over 5 years | 16 | 70% |
| 2–5 years | 6 | 26% |
| Up to 1 year | 1 | 4% |
| Ever received research ethics-related training | | |
| Yes | 20 | 94% |
| No | 2 | 3% |
| No response | 2 | 3% |
| Ever reviewed research protocol that included broad consent for future use of data or samples | | |
| Yes | 19 | 79% |
| No | 5 | 21% |
| No response | 0 | |
| Ever approved research protocol that included broad consent for future use of data or samples | | |
| Yes | 15 | 62% |
| No | 4 | 17% |
| No response | 5 | 21% |

(*Continued*)

**Table 1.** (Continued)

| | N | % |
|---|---|---|
| Guidelines followed by ERC allow for acceptance of broad consent for future use | | |
| Yes | 12 | 50% |
| No | 9 | 38% |
| No response | 3 | 12% |
| ERC provides guidance on how to review protocols that include future use | | |
| Yes | 13 | 54% |
| No | 9 | 38% |
| No response | 8 | 8% |
| ERC guidance for how to review protocols that include future use includes additional guidance for future use considerations in epidemics and pandemics | | |
| Yes | 4 | 17% |
| No | 16 | 67% |
| No response | 2 | 8% |
| Not applicable (no guidance from ERC) | 2 | 8% |

ERC = Ethics Review Committee.

*"In some cases, there is enough expertise within the committee to make it easy to deal with [use of future samples]. There are some cases where the expertise is very limited and only one or two people handle it, in some cases no one handles it."*

*(CEI-010, male, Chairperson, over 45 yrs. old)*

## Heterogeneity in interpretation of broad consent for future use

ERC members with more experience or advanced training on research ethics felt comfortable describing the concept of broad consent for future use. A few participants described broad consent for future use as the reuse by the same investigators or for the same purpose for which the data or samples were originally collected and some participants did not feel comfortable explaining the concept of broad consent for future use. Participants also described a need for further training to clarify the definition and limits of broad consent for future use.

*"I am not very clear about what broad informed consent refers to, if it is a general consent wherein the characteristics of the project are not specified in great detail and like the individual's duties and rights or if it refers to asking the individual for the subsequent use of [their data] for other research projects."*

*(CEI-012, Female, Chairperson, 35–45 yrs. old)*

ERC members expressed concerns that broad consent for future use could be taken, instead, as a blanket consent. Several members compared broad consent for future use to giving researchers a blank check or a backdoor.

*"Something that I would not approve of is to leave the broad informed consent like a blank check, something that remains like. . .open."*

*(CEI-004, male, Chairperson, 35–45 yrs. old)*

ERC members emphasized the need to limit the range of future uses, either by pre-specifying those uses in the research protocol or having another mechanism for ERC members to evaluate the research questions related to data or sample reuse. These concerns were magnified when considering sample sharing where there was more concern related to exploitative research, including for profit research that did not benefit the source community.

*"Biorepositories are used as a source, as a backdoor to do certain things"*

*(CEI-010, male, Chairperson, over 45 yrs. old).*

## Balancing concerns of different stakeholders

Participants discussed how they weigh the risk to individual research participants against the potential benefits to society. Central to this assessment was the understanding that data and sample sharing should benefit society.

*"It is possible to make use of that information, of that sample, for the benefit of society, of humanity. To make that possible one has to have an altruistic sense of the purpose for the use of those samples."*

*(CEI-009, male, Chairperson, over 45 yrs. old)*

While some participants said that anonymization provided adequate protection for research participants,

*"If the data are correctly anonymized there should be no risk. . .there should be no risk of identification of the person."*

*(CEI-003, male, President, 35–45 yrs. old)*

Other participants highlighted the concern that samples and their associated data could not be completely anonymized.

*"The risk is that [the biological samples] are taken from. . .research and that they are related to health, and there is always a way to identify the participant"*

*(CEI-014, female, Chairperson, over 45 yrs. old)*

Participants stressed that societal benefit should not trump individual rights, including the need to ensure that participants understand the implications of broad consent for future use.

*"The social benefit of research is important but. . .cannot corrupt individual rights. . .They are not corrupted when the [researcher] has described the information from the outset in such a way that the subject has the information and wants to give his sample."*

*(CEI-010, male, Chairperson, over 45 yrs. old)*

When considering the inclusion of broad consent for future use in research protocols that plan to work with indigenous or other vulnerable communities, ERC members described carefully weighing risks to those communities against a benefit to the larger community with the suggestion that researchers consider the reuse of data from non-vulnerable populations before sharing data or samples from a vulnerable population.

*"If these samples come from vulnerable communities, let's say if they come from people in conditions of poverty, people from vulnerable ethnic or racial groups, in that case we take a critical view, we question. . .What is the role of this type of request? Why do they want to extend consent? Why don't they work with what they have. . .or to what extent did they request consent?"*

*(CEI-013, male, Chairperson, over 45 yrs. old)*

ERC members indicated they are more likely to approve both broad consent for future use and waivers of consent in research protocols that demonstrate a clear potential for public health benefits related to reuse, sufficient protections for participants in the form of appropriate anonymization and protections for sample maintenance and transfer, and the assurance that data or sample reuse will be limited to academic rather than open to commercial partners.

The ERC members interviewed highlighted sharing data or samples with the pharmaceutical industry as a central concern when evaluating research projects that include broad informed consent for the future use of data and samples.

*"So regardless of whether it's a sponsor who might have an economic interest or a public health institution, above all, industry, can misuse [the samples]. Why would they want [the samples]? They are going to get a product, and whoever's contributing their samples will never get any benefit, right?"*

*(CEI-001, female, Vice President, over 45 yrs. old)*

Researchers felt that academic partners would be more likely to apply good governance, including ethical and equitable data and sample storage, management and reuse, than for profit companies, including the pharmaceutical industry.

*"In research studies at the educational level, at the university level, it is healthier. At the industry level, because political and financial interests are involved, it lends itself to any deviation, to any abuse"*

*(CEI-008, male, community representative, over 45 yrs. old)*

ERC members mentioned struggling to ensure that all voices on the ERC were considered in the evaluation of broad consent within the research protocol. Several people reported feeling that the group could subsume individual voices.

*"I believe that the problem with many committees is that they erase the voice of the other [ERC member], and that is why there is offense. There is offense because the researcher feels ignored, using this colloquial expression, well. . .or degraded."*

*(CEI-017, male, President, over 45 yrs. old)*

## Community involvement in the ERC

In Colombia, ERCs are required to include a member of the community to ensure the inclusion of the community's inputs in the research review process. That said, the community members that serve on ERCs said they limit their review and comments to the informed consent form rather than the full research protocol and do not participate in the conversations related to the protocol review.

*"As a representative of the community, I must read an informed consent. . .as I do not belong to the health field, I must understand it, the words, the context, everything. . .so, I do not review the entire protocol, I only read the consents."*

*(CEI-007, female, community representative, over 45 yrs. old)*

*"Almost always they ask me if I understood the consent."*

*(CEI-019, male, Chairperson, 35–45 yrs. old)*

Several participants mentioned the importance of opening a space for the community to engage with how their data and samples are driving scientific or practice-related advances. They said that, in most cases, broad informed consent is proposed by researchers without the involvement of the community or civil society.

*"We have to create spaces to educate the community and people in research. This pandemic has exposed the need to do research, has shown. . .in our environment the little knowledge, and the need to educate and also, the interest that people. . .and. . .communities can have in knowing about. . .what research is, research ethics, the processes that are in all this, the informed consent processes, etc. So, I believe that people should absolutely be involved in an educational process that allows them to participate in the decision-making process related to the generation of knowledge related to how they are, their lives, the problems that they have, the resources they provide, like their biological samples."*

*(CEI-001, female, Vice President, over 45 yrs. old)*

*"We used to do little [projects], very fragmented like a research ant operation, and those projects accumulated and we were left with a body of knowledge. Now, the research enterprise has changed. We know that serious research will be structured, require a lot of time, will go in sequence, will be accompanied by these communities, with work with these communities. I think our researchers' thinking has matured to include a larger picture."*

*(CEI-017, male, President, over 45 yrs. old)*

*"The researchers are the ones who propose this broad informed consent. We have never seen the participation of the community or civil society, for example, their involvement in these projects. Research projects that are presented for discussion [by the ERC] are almost never suggested by the community or by anyone other than the investigators themselves. That should change too, right? From the proposal of the research objectives, people should be given a bigger role, common people, not just the investigators themselves."*

*(CEI-012, female, Chairperson, 35–45 yrs. old)*

### Changes in ERC procedures in response to COVID-19 pandemic research

Most participants reported that their committee had modified standard procedures in response to COVID-19, including expedited review and steps to facilitate data sharing.

*"I think that in the bureaucratization of the investigation and of the ethics committees we became pachydermic. . .we delayed for a time that. . .should not be necessary. The pandemic*

*has shown that. . .we can be more efficient, and that we can evaluate in 3 days. . .we can. . .have approvals in that period of time."*

*(CEI-001, female, Vice President, over 45 yrs. old)*

## Risks of broad consent for future use

ERC members described risks as relating to participants not understanding the implications of broad consent for future use, the failure of groups that reuse the data to preserve the confidentiality of data, especially genetic data, and the identification of incidental findings without then communicating these findings to individual participants.

*"What if they manage to identify a clinical alteration in those samples?. . .well. . .allow broad consent, but. . .if any inherent condition is found in the samples, ask the patient if they want to know that inherent information from their samples. . .It is this total dissociation of the relationship of samples versus research, which. . .is that the patient ends up being an input, another provider of information, there is no interaction or benefit. . .that can derive for him."*

*(CEI-004, male, Chairperson, 35–45 yrs. old)*

Risks for investigators included parachute research, where researchers were not engaged in studies that use their data and the benefits of data or sample reuse do not return to the research participants' community. The use of samples from Colombia to develop COVID-19 vaccines that could not be used in Colombia because of the lack of appropriate refrigeration was cited as a concern.

*"Here come the researchers for the COVID vaccine. . .which must be kept at minus 70 degrees. . .Here, where do we have freezing temperatures at minus 70?"*

*(CEI-001, female, Vice President, over 45 yrs. old)*

Lastly, participants described the potential for future studies to identify characteristics of a given community associated with the utility of a given treatment or the propensity to a certain disease which could then stigmatize that community and/or limit their access to treatment or insurance coverage.

*"These studies can generate pharmacogenetic profiles, which makes communities easily identifiable later so that they are, let's say, conveniently. . .identified for certain drugs and others not, that happens, that there are, let's say, certain biomarkers for disease susceptibility, and even though these people never consented to that, they were then identified, which makes them vulnerable, and let's say that, obviously, if these databases are sold or the confidentiality of the data is lost, well, these communities are susceptible to potential discrimination based on those genetic markers, right? Or, for example, the research participant is susceptible to the eventuality that they identify that they have a neurological disease and they could then be discriminated against at work. . .when I never consented to all of this they ended up pointing me out, like a nuclear reaction, all of these studies can identify things that I don't know about and that I didn't consent to and that can have consequences later for insurance, access to prepayment [for medical care], access to jobs, because you have been identified as a person who is susceptible to X or Y medicine. So yes, there are real situations where one is left vulnerable, exposed."*

*(CEI-020, male, President, 35–45 years old)*

## Need for well defined, good governance

ERC members expressed concern related to the international transfer of biospecimens, including the need for clearly defined, transparent governance to ensure ethical and equitable sample management.

> *"Who safeguards the information? Who is in charge of defining whether this sample leaves or does not leave for investigation, and. . . the person directly affected, then, to administer and safeguard their information? Another [point] is a defined area. . . a boundary, so to speak, to that expanded consent, yes? At least a subject area, or at least a defined disease, or at least a defined context, not something that remains open."*

> *(CEI-004, male, Chairperson, 35–45 yrs. old)*

ERC members discussed the need to look downstream to understand what would be done with the research results, to whom they would apply or be communicated.

> *"The researcher's [interest] is to be able to produce and build their work and reputation. Another thing is to look at the institutions involved, that is, who are the parties involved and what do they want to do with the results, that is, where do these results go, to whom they are addressed"*

> *(CEI-005, male, Chairperson, 35–45 yrs. old)*

ERC members specified that biobank governance should include: mechanisms to fulfill requests to withdraw or destroy their samples at any time; quality control for sample handling and analysis; clearly specified steps to ensure the secure transfer and storage of samples and the confidentiality of related data, with a focus on preserving the confidentiality of genetic material. The reuse of human samples for use by for-profit, commercial entities was flagged as a concern both because of lack of benefit sharing and because ERC members felt that the community would not agree with companies profiting from their samples.

At a minimum, ERC members expected prespecified inter-institutional agreements on the use of human specimens by for profit companies. ERC members expressed concern that there was no way to know whether biobanks had met these requirements as the biobank governance structure is not specified in the protocol and oftentimes the investigators who have submitted the protocol are not familiar with the biobank's governance structure.

> *"There must be an established data governance. The samples will belong, according to the protocol, to those who have stipulated the governance, which in most cases is the research group or the institution. . .The institution must act as guarantor and protector of that information, because behind that information there is the researchers' academic work and there is a. . .biological remnant and an input provided by the patients. So, although it belongs to them. . .it does not mean that they can do whatever they want with the data, but that they must participate as guarantor and. . .protector of what is derived from those samples or those data."*

> *(CEI-004, male, Chairperson, 35–45 years old)*

Participants specified that they would feel differently if core components of the governance of data and sample sharing were clearly specified in the research protocol, including which groups or individuals are overseeing the sharing process, how long end users will keep data or

samples, and how groups that reuse data or samples will destroy the material or information when the work is completed.

*"I think it is not written down as much as I wish it were. . .Let's say, that the protocol complies with the standard, it complies with a norm that says, yes. . .it is that they identify the place where the samples will be stored, the biobanks, the repositories, where they are, in what city. But it is not very clear. . . I would think that. . .greater emphasis should be placed on that, on that. . .the governance."*

*(CEI-014, female, Chairperson, over 45 yrs. old)*

In contrast to poorly specified governance, ERC members described the benefits of well structured, prospectively described governance as enabling ERC members to view the future use of data and samples with optimism, support intra- and inter-institutional collaboration, and facilitate important scientific advances.

*"Precisely the figure of a strengthened, well-regulated biobank, with good will. . .it makes the objective of scientific cooperation more important."*

*(CEI-017, male, President, over 45 yrs. old)*

## Fear of coercive or uninformed consent

Participants were concerned that researchers did not take sufficient steps to ensure that participants understood what broad consent for future use means and does not mean and specified that researchers might deliberately take advantage of participants' lack of understanding to secure a wide spectrum of potential future uses.

*"The [Trojan] horse within the ethics committee is that the consents are not sufficiently clear, so that a person with the level of education that the patient is expected to have can read the consent and know what is happening. . . People's rights must be respected, society must respect the individual and. . .to respect them, consents must be well formulated."*

*(CEI-010, male, Chairperson, over 45 yrs. old)*

*"So. . .what is a competent individual? An individual who is physically and morally. . .in his right mind. But I would add something else, the cultural, because it is very easy and you see it. . .to build a consent where you do not understand anything, and in some cases the person almost doesn't know how to read or write and they ask him, if he is in agreement, to sign there and. . .the problem that then ensues is that of free decision."*

*(CEI-009, male, Chairperson, over 45 yrs. old)*

*"When you signed, which was what I explained to you (. . .) when you signed that consent, it was not very clear to the patient how far, right? the scope, and there may be a risk that he would not want that research or the result or his data to be used beyond what he initially understood could be used, right?"*

*(CEI-023, female, community representative, over 45 yrs. old)*

To address the need for participant comprehension, ERC members suggested using clear, precise, and concise language, avoiding technical terms, and providing a clear explanation

regarding how de-identification can be guaranteed. They also suggested the need to ensure that the informed consent document is explained by trained research staff that the participant feels is approachable and who can respond to the participant's questions.

*"A risk is, sometimes the lack of knowledge of the investigation processes, which leads them to a low level of understanding of the informed consents. It is not something new that when research is conducted, you have to really try to ensure that the informed consent is clear to the participants. And sometimes, in this context of broad consent, it ends up depending on good faith or on the participant's hope for the investigator's good faith."*

*(CEI-004, male, Chairperson, 35–45 yrs. old)*

## Collaboration between high and low-and-middle-income countries

ERC members described both benefits and risks associated with sharing data or samples with investigators in high-income countries. One respondent cited this exchange as a way to maximize the utility of the data or samples by applying new technologies that are not yet available within Colombia.

*"Technologies that exist in that country or in that institution are not available, so one has to work with other institutions."*

*(CEI-016, female, President, over 45 yrs. old)*

Another respondent expressed concern that sample sharing, in particular, could lead to extractive research with no clear benefit for the source community.

*"We also want to avoid this use of samples in Latin America to do extractive research. That is to say, Latin America as a sample dispenser only. . .only as a collector."*

*(CEI-017, male, President, over 45 yrs. old)*

## Benefits of broad consent for future use

Participants felt that data reuse could maximize the utility of data or samples and lead to new collaborations and capacity building for researchers and discoveries at the societal level.

*"It has many benefits, identification of diseases linked to genes associated with the genotype of that community, effects of drugs, effects of a vaccine or effects of an infectious or non-infectious disease on that type of genetics or genome."*

*(CEI-003, male, President, 35–45 yrs. old)*

Several participants mentioned that data and sample sharing had benefitted COVID-19 response by enabling the rapid development of diagnostics and vaccines.

*"In cases of infections, pandemics like now, we are sure that vaccines could not have been developed so quickly, if there were not biobanks and data repositories available to do so, then the advantage is that there is information, there is enough to investigate and work because, in what is required, it is available, that is the greatest benefit."*

*(CEI- 014, female, Chairperson, over 45 yrs. old)*

*"I see it as very pertinent precisely because having broad consent allows us to have a good use. . .well. . .of resources. Especially in this country where it is so difficult to get resources for research, and that obtaining the samples takes a good percentage of the research money."*

*(CEI-026, female, Chairperson, over 45 yrs. old)*

*"There is a benefit for. . .the cure of future diseases, if we are talking about disease or for future generations as well. And on the other hand, there is the benefit of training new researchers in these topics, indirectly it would also benefit the communities because we would have more researchers properly trained."*

*(CEI-016, female, President, over 45 yrs. old)*

## Data as a global public good

Participants spoke about data as a global public good, including the idea that anonymized data should be open and novel analytic approaches to big data meant that studies that were not validated or confirmed by additional research had effectively been replaced by global analyses.

*"The sample that is taken from me today with an identification, etc., etc., well, it belongs to me, that sample will allow for the generation of a large amount of information and data, which, together with other samples will produce research. . .and that, well. . .it belongs to him or it belongs to us. Whatever is generated from that, let's say it like this, it belongs to everyone. Yes? To all those who participated, to those who participated in the generation of the project, and in general to humanity because it is the generation of knowledge that should, which must be universal and global and accessible to all."*

*(CEI-001, female, Vice President, over 45 yrs. old)*

*"When the data are anonymized—then they belong to the Open Science community."*

*(CEI-009, male, Chairperson, over 45 yrs. old)*

*"We can no longer think that an isolated research group makes a discovery and with that validates everything. . .we are really in the world of big data. Biostatistics is showing us that many times we are more effective when we are properly united and this will imply a homologation."*

*(CEI-017, male, President, over 45 yrs. old)*

Participants also related the consideration of data as a global public good to the COVID-19 pandemic, suggesting that a global public health emergency necessitates global collaboration.

*"And I believe that this pandemic has put it in our faces, it is not local, not regional, not mine, nor is the knowledge mine, but rather belongs to everyone, we all have to contribute, we all have to put it, it costs us all. Yes? I think that, in [the research] field, that is what we have learned from the pandemic."*

*(CEI-01, female, Vice President, over 45 yrs. old)*

In Fig 1 and S1 Table, we summarize these findings into facilitators for and positive effects of broad consent for future use and, conversely, concerns about broad consent for future use and the possible negative effects of that use. S1 Table also includes the original Spanish language for the quotes presented above.

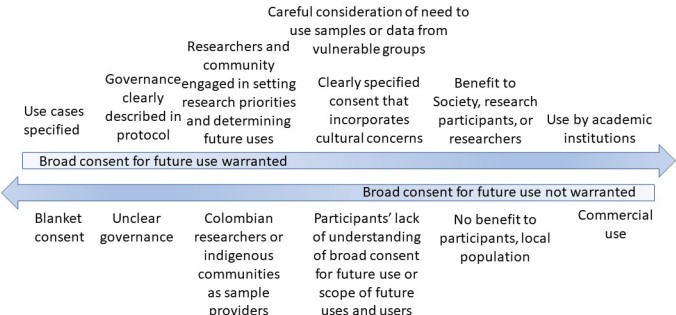

**Fig 1. Facilitators and barriers related to ethics review committee members consideration of broad consent for future use of data and samples in research protocols.**

## Discussion

In this cross-sectional, qualitative study, we explored how ERC members in Colombia weigh risks and benefits related to data and sample reuse and the rights of different stakeholders when reviewing protocols that include broad consent for future use or applications for waivers of consent in the absence of national policies or laws related to data and sample sharing. We did not find meaningful differences in how male and female ERC members described concerns about broad consent for future use but did observe that participants who held leadership positions on the ERC and had more experience on ERCs felt more comfortable describing broad consent for future use and were more likely to highlight the benefits of data reuse than ERC members with less experience. While the required inclusion of community members on ERCs in Colombia is an important step to ensure community representation, their role could be expanded beyond review of the informed consent form and additional forms of community engagement in setting data or sample reuse priorities would help ensure meaningful benefit sharing. ERC members were uncomfortable with the lack of information on how data and sample sharing would be managed, suggesting the need for pre-specifying the governance framework for data and sample sharing in the research protocol. The lack of national guidelines or laws related to data and sample reuse was cited as a concern and likely adds to the perception of protocols that include broad consent for data or sample reuse as representing a higher risk for research participants. These risks were especially important for ERC members considering protocols that included broad consent for future use of data and samples from vulnerable populations. ERC members reflected on how COVID-19 has changed the approach to understanding who owns the data or how the data can be used, including highlighting the utility of biobanks in fast-tracking vaccine development and the need for collaboration in response to a global pandemic.

The pre-specification of reuse purposes, clarification of the governance framework, ensuring participants understood broad consent for future use-related language in the informed consent form, and engaging of the community in setting public health priorities for data reuse were cited as ways to assure ERC members of ethical and equitable application of broad consent for future use. While data and sample sharing between high and LMIC was seen as a way to foster collaboration, parachute research and exploitation, particularly the reuse of samples for the development of COVID-19 vaccines that would not benefit Colombia, were highlighted as important concerns and related to when the research was conducted, between February and May of 2021 when COVID-19 vaccines were widely available in some high-income countries but not in LMIC, including Colombia. ERC members described the COVID-19 pandemic as

changing ERC review processes, including fast-tracking protocol review and the conceptualization of the research enterprise and data and sample sharing as global endeavors rather than existing within a limited network or one-sided relationship.

ERC members had trouble conceptualizing themselves as research subjects and wanted to review the research protocol and interview guide before the interview. While investigators assured participants that there were no right or wrong answers, the inclusion of a member of the Ministry of Science and Technology as an interviewer may have led to the decision by some ERCs to not participate in the study. ERCs that opted not to participate may have felt more uncomfortable with the concept of broad consent for future use than those that agreed to participate in the study.

The concerns raised by ERC members in this study reflect those of research participants and experts in research ethics presented in other studies and reviews. For example, ethicists have highlighted the tension between legal and normative frameworks for data sharing, including the GDPR requirement that consent is specific and granular which can imply the need to pre-specify the range of future uses [18]. ERC members expressed the need to limit the reuse of data or samples for commercial purposes, which reflects the preferences of research participants in other studies [19–21] and stakeholders engaged in biobanking oversight or management [22]. ERC members' concerns about the risk of extractive research, which fails to build within-country research capacity, when sharing data or samples with high income countries was similarly cited as a concern in another qualitative study with 11 ERC members in Malawi [23]. Also similar to other studies, ERC members highlighted communication of incidental findings as a potential benefit to biobanking participants [24].

Similar to what ERC members reported in this study, indigenous and other groups have highlighted the need to account for cultural considerations and the concern that sample sharing could lead to further targeting of already vulnerable groups [25]. The concern that reuse of data or samples from indigenous or otherwise marginal groups should be very clearly justified was also reflected in an earlier study wherein IRB members reflected on the consequences of the Havasupai Case for consideration of broad consent for future use in research studies that involve participants from vulnerable ethnic or racial groups [26]. Challenges related to the lack of national guidance or laws regarding broad consent for future use [27] and the need to differentiate broad from blanket consent [28] have also been identified in prior studies or meeting reports. At the same time, international ethics bodies have called for the prospective consideration of broad consent for future use of data and samples during public health emergencies [29,30]. The role of ERCs in the governance of future use is an active area of discussion [31,32]. There is a clear need for guidelines on governance that can be used by ERCs to evaluate whether basic expectations for best practice will be met. While the interview guide included questions related to both data and sample sharing, ERC members tended to focus their discussion on sample sharing-related concerns which highlights the importance of clear governance and community oversight for biobanking initiatives and suggests that sharing participant-level data may be less of a concern to stakeholder groups or subject to different processes for weighing related risks and benefits. Similar to the findings from a stakeholder meeting with ERC members in Africa, ERC members in Colombia were supportive of the concept of broad consent for future use of data and samples given transparent governance that fairly weights community and research participant concerns and benefits [28]. Colombian ERC members suggested the pre-specification of governance procedures and biobanking or data management facilities in the research protocol as a way to address concerns about the management of future uses.

ERCs' chief role is to protect the rights of research participants and ensure the social value of research through evaluating research quality and weighing the potential benefits and harms

associated with research. Multicenter trials and cross-national research present ethical, legal, and logistic challenges for investigators and ERCs in LMIC countries [11,33]. Many LMICs, including Colombia, do not have national legislation or clear policies related to broad consent for future use for data and samples [11], including sharing data or samples with other countries. The key ethical principles of justice, autonomy, beneficence, and nonmaleficence may be interpreted differently in different contexts [34]. For example, the intervention [35] and protection [36] models of bioethics, which privilege the role of shared (public) health concerns over individual rights and consider health justice in the presence of scarcity with a focus on vulnerable populations [37], emerged from LAC [38] and can be contrasted with principalism, which emerged from Europe and is rooted in individual rights [39].

Primary research on how ERCs evaluate human subjects research applications that include broad consent for future use is extremely limited. We searched Ovid(Medline) from 1 January 2000 to 21 November 2021 using the following title and abstract text terms: ((future use* OR (broad OR blanket OR wide OR open OR data shar*) adj2 (consent OR data shar*)) AND ((ethics adj2 committee* OR 3 institutional review board* OR ethics committee* OR research ethics OR ethics review)) to identify existing primary research related to ERC members consideration of broad consent for future use. We only identified two qualitative research studies that explored broad consent-related knowledge, attitudes, or practices of ERC members in Malawi (N = 11) [23] and the US (N = 3) [40] and a meeting report from ERCs in 15 African countries (N = 41) [28]. Concerns expressed by ERC members in Colombia in this study reflect the findings of these two studies and the meeting report. While the similarities across these three small studies cannot be the basis for any type of generalization, they do suggest a convergence of ERC members' attitudes towards broad consent for future use that should be further explored and addressed. As the first study to address how Colombian ERCs weigh different stakeholders' concerns when reviewing protocols that include broad informed consent, and one of only two other research studies on the topic in any country, provides insight into the challenges faced by ERC members when evaluating provisions in the IC that facilitate future collaborations between countries, including between developed and developing countries.

In the absence of national regulation, LMIC may default to laws and policies enacted in different countries or regions, including the GDPR, data sharing legislation enacted in Europe, and the CIOMS guidance [8], which includes considerations for broad consent for future use. Understanding how ERCs make decisions about future use in the absence of national legislation is important for understanding how ERC members weigh the considerations of different stakeholders when making decisions about broad consent for future use and has been flagged as a concern by other groups [41]. Even when there are national laws or policies, some studies indicate that ERC members' decision making may not correspond to those [42,43]. While the presence of national legislation doesn't necessarily mean that ERC members' evaluations of broad consent for future use in similar protocols for similar purposes will be aligned, encoding national values and preferences would likely help with such alignment. Given ERC members' importance as research gatekeepers, future research is needed to understand ERC decision making processes around broad consent for future use regardless of the presence of national legislation.

## Conclusion

ERC members represent an understudied and pivotal stakeholder group in the data and sample sharing ecosystem. ERCs moderate research participant and public trust in research. Understanding how ERC members evaluate the risks and benefits of data and sample sharing

can help researchers, funders, and the open science community understand how to better address outstanding concerns related to data and sample reuse. While some researchers and policymakers have called for standardized approaches to the evaluation of broad consent for future use [44,45] others have emphasized the need for context- and use-case specific approaches. Previous research has highlighted the need to understand how ERCs make decisions about broad consent for future use, including the development of a pilot tool for documenting the decision-making process [41]. Concerns expressed by ERC members in Colombia reflect those of research participants in diverse contexts and from ERC members in Africa [23] and the US [40]. Understanding how ERC members perceive broad consent for future use and their perception of the harms and benefits of data sharing can help researchers better address these concerns. Future research is needed to explore how to document ERC decision making processes around broad consent for future use and their correspondence with community values.

ERC members described the continuum along which they evaluate broad consent for future use where data and sample sharing can be both a Trojan horse and a global public good. The steps that researchers take to specify the governance structure and organizations who will manage future data and/or sample sharing, how researchers engage the community to determine preferences for future use, how well ERC members think participants will understand the subject information sheet and informed consent, future use by commercial entities, etc. all help determine how broad consent is perceived by ERC members. Several frameworks for ethics review in public health emergencies have highlighted the need for ERCs to address data and sample sharing [29]. ERC members' concerns reflect a subsequent call from the WHO to align and support equitable governance structures to realize health research data as a global public good [46]. As such, study findings have implications for researchers and ERCs in Colombia and in other countries that do not have a legislative framework for broad consent for future use or data and sample sharing during the ongoing COVID-19 pandemic and beyond.

## Supporting information

**S1 Table. Representative quotes and corresponding themes.**
(PDF)

**S1 Text. PLOS inclusivity in global health questionnaire.**
(PDF)

**S2 Text. Semi-structured in-depth interview guide.**
(PDF)

## Acknowledgments

The authors would like to thank the ERC members who participated in the study for their important contributions to this work and Drs. Elena Rey, Norma Serrano, Martha Moyano, Abha Saxena, Susan Bull, and Calvin Ho for their feedback on the objectives and design of this research.

## Author Contributions

**Conceptualization:** María Consuelo Miranda Montoya.

**Formal analysis:** María Consuelo Miranda Montoya, Jackeline Bravo Chamorro, Luz Marina Leegstra, Deyanira Duque Ortiz, Lauren Maxwell.

**Funding acquisition:** Lauren Maxwell.

**Investigation:** Jackeline Bravo Chamorro, Deyanira Duque Ortiz.

**Methodology:** Jackeline Bravo Chamorro, Luz Marina Leegstra, Lauren Maxwell.

**Project administration:** María Consuelo Miranda Montoya, Lauren Maxwell.

**Supervision:** María Consuelo Miranda Montoya, Luz Marina Leegstra, Lauren Maxwell.

**Writing – original draft:** Lauren Maxwell.

**Writing – review & editing:** María Consuelo Miranda Montoya, Jackeline Bravo Chamorro, Luz Marina Leegstra, Deyanira Duque Ortiz, Lauren Maxwell.

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
