## [Decision Letter · Decision Letter 0]

20 Jan 2022

PGPH-D-21-01141

A blank check or a global good? A qualitative study of how ethics review committee members in Colombia weigh the risks and benefits of broad consent for data and sample sharing during a pandemic

Dear Dr. Maxwell,

Thank you for submitting your manuscript to PLOS Global Public Health. After careful consideration, we feel that it has merit but does not fully meet PLOS Global Public Health’s publication criteria as it currently stands. Therefore, we invite you to submit a revised version of the manuscript that addresses the points raised during the review process.

The reviews are on the whole positive, but a few key issues have been raised and I think dealing with them will help your manuscript reach it's full potential. I'm glad that you have referenced the Standards for Reporting Qualitative Research - but there are a few points in these guidlelines that you could pay closer attention to. In particular it is not clear if there was any particular qualitative approach or guiding theory for this work. There could be more detail in the description of the qualitative analysis and could you reference a particular paradigm or process? I'm interested in whether you used any techniques to enhance the trustworthiness and credibility of the analysis, including member checking.

We look forward to receiving your revised manuscript.

Kind regards,

M. Dylan Bould

Academic Editor

Journal Requirements:

2. In the online submission form, you indicated that "The study protocol, demographic information sheet, and IDI guide are available on the Open Science Foundation (10.17605/OSF.IO/2YUVF). Relevant study data are presented in the manuscript text and tables. Due to the descriptive nature of the data and the small sample size, further data will not be made available to protect participant confidentiality.". 

All PLOS journals now require all data underlying the findings described in their manuscript to be freely available to other researchers, either 1. In a public repository (working direct link), 2. Within the manuscript itself, or 3. Uploaded as supplementary information.

Additional Editor Comments (if provided):

Reviewers' comments:

Reviewer's Responses to Questions

**Comments to the Author**

1. Does this manuscript meet PLOS Global Public Health’s publication criteria? Is the manuscript technically sound, and do the data support the conclusions? The manuscript must describe methodologically and ethically rigorous research with conclusions that are appropriately drawn based on the data presented.

Reviewer #1: Yes

Reviewer #2: Yes

Reviewer #3: Yes

2. Has the statistical analysis been performed appropriately and rigorously?

Reviewer #1: N/A

Reviewer #2: Yes

Reviewer #3: N/A

3. Have the authors made all data underlying the findings in their manuscript fully available (please refer to the Data Availability Statement at the start of the manuscript PDF file)?

Reviewer #1: Yes

Reviewer #2: Yes

Reviewer #3: Yes

4. Is the manuscript presented in an intelligible fashion and written in standard English?

Reviewer #1: Yes

Reviewer #2: Yes

Reviewer #3: Yes

5. Review Comments to the Author

Reviewer #1: This is a good article that addresses a topic of the utmost importance that needs to be urgently regulated by the States. In this sense, the research offers an inventory of aspects that need to be taken into account in the elaboration of this regulation. It is also very important that qualitative studies are valued and that articles on issues related to research ethics are encouraged. Another positive aspect of the text is to give voice to members of ethics committees, which is very rare in the literature. The question elaborated in the title corresponds to a good research problem. The qualitative research seems well planned and executed.

However, some doubts remain. The first one would be: why Colombia? Since Plos Global Public Health is an international publication, it would be important to explain to readers whether Colombia is a country that can be considered a standard or an exception regarding the legal treatment of this issue, but also regarding the work of the ERC and scientific research. Do most States have rules or not? Is the fact that Colombia is a developing country related to the lack of regulation of the matter pointed out by the authors? In other words, contextualizing the situation of the chosen country in the global scenario would contribute to a better use of the results.

Regarding the opinion of ERC members, at the end of the text the authors write: "Similar to the findings from a stakeholder meeting with ERC members in Africa, ERC members in Colombia were supportive of the concept of broad consent for future use of data and samples given transparent governance that fairly weights community and research participant concerns and benefits [20]"(603-606) and "Concerns expressed by ERC members in Colombia reflect those of research participants in diverse contexts and from ERC members in Africa and the US" (618-619). This point which would be one of the major contributions of the paper, but was the subject of only two sentences that are based on only one reference. The idea is floated but not developed.

Also missing is the reference to other articles that address exactly the main issue of the research during an international emergency - for example Alirol, E., Kuesel, A. C., Guraiib, M. M., de la Fuente-Núñez, V., Saxena, A., & Gomes, M. F. (2017). Ethics review of studies during public health emergencies - the experience of the WHO ethics review committee during the Ebola virus disease epidemic. BMC medical ethics, 18(1), 43. https://doi.org/10.1186/s12910-017-0201-1.

In summary, I believe it is essential that the article devotes a few paragraphs to the global context, both with regard to national standards and with regard to the views of ERC members from other countries, as this literature already exists and can be consulted.

Finally, I believe that the question formulated in the title of the article is not sufficiently answered in the final considerations which could be improved.

Reviewer #2: In this cross-sectional qualitative study using a semi-structured interview of a broad range of members of Ethical Review Boards in Colombia, authors circumscribed the research question of the perception of the ERCs’ members on the broad consent for future use.

Perceived risks and potential benefits at research participant-level, community level and to more extent benefits and risks when sharing data or samples are operated between LMICs and high income countries including data used for profit purposes are explored. The interviews did not omit to include the perception of this issue with regard to the current COVID-19 pandemic.

This study is of great interest as it raises concerns surrounding this topic and draws some potential orientation for solutions including the governance and need to broaden information on possible use of future researches of data or samples in order to obtain an authentic informed consent for participants and guarantee of data safeguarding.

I hereby identified some minor comments:

- Line 100: …related to instead of…..relate to…

- Line 120: ….interventional instead of …. intervention

- Table 1: use Male and Female …..instead of….. Masculine and Feminine

Occupation may rather be replaced by Professional background

- Line 202: ..selected protocols instead of select protocols

Lines 114-116: The research team reviewed related literature and consulted with researchers who work on data and sample sharing in a number of different LMICs and globally.

The role or impact of this review on this study should be well mentioned in Methods

Lines 138-150: around the data collection (interviewers) especially the philosopher… If these researchers are among the authors for this manuscript, the initials of their names should be mentioned in parentheses to ease the riders' work.

Thank you for good and commendable piece of work

Reviewer #3: Excellent paper. It addresses an important emerging area of policy and practice and the analysis is well supported by the research. This will surely make a substantive contribution to inform and further the development of research ethics standards and guidelines for broad consent for future uses of data.

6. PLOS authors have the option to publish the peer review history of their article (what does this mean?). If published, this will include your full peer review and any attached files.

**Do you want your identity to be public for this peer review?** For information about this choice, including consent withdrawal, please see our Privacy Policy.

Reviewer #1: No

Reviewer #2: No

Reviewer #3: No

---

## [Decision Letter · Decision Letter 1]

13 May 2022

A blank check or a global public good? A qualitative study of how ethics review committee members in Colombia weigh the risks and benefits of broad consent for data and sample sharing during a pandemic

PGPH-D-21-01141R1

Dear Dr. Maxwell,

We are pleased to inform you that your manuscript 'A blank check or a global public good? A qualitative study of how ethics review committee members in Colombia weigh the risks and benefits of broad consent for data and sample sharing during a pandemic' has been provisionally accepted for publication in PLOS Global Public Health.

Best regards,

M. Dylan Bould

Academic Editor

Reviewer Comments (if any, and for reference):

Reviewer's Responses to Questions

**Comments to the Author**

1. If the authors have adequately addressed your comments raised in a previous round of review and you feel that this manuscript is now acceptable for publication, you may indicate that here to bypass the “Comments to the Author” section, enter your conflict of interest statement in the “Confidential to Editor” section, and submit your "Accept" recommendation.

Reviewer #1: All comments have been addressed

Reviewer #2: All comments have been addressed

Reviewer #3: All comments have been addressed

2. Does this manuscript meet PLOS Global Public Health’s publication criteria? Is the manuscript technically sound, and do the data support the conclusions? The manuscript must describe methodologically and ethically rigorous research with conclusions that are appropriately drawn based on the data presented.

Reviewer #1: Yes

Reviewer #2: Yes

Reviewer #3: Yes

3. Has the statistical analysis been performed appropriately and rigorously?

Reviewer #1: N/A

Reviewer #2: Yes

Reviewer #3: Yes

4. Have the authors made all data underlying the findings in their manuscript fully available (please refer to the Data Availability Statement at the start of the manuscript PDF file)?

Reviewer #1: Yes

Reviewer #2: Yes

Reviewer #3: (No Response)

5. Is the manuscript presented in an intelligible fashion and written in standard English?

Reviewer #1: Yes

Reviewer #2: Yes

Reviewer #3: Yes

6. Review Comments to the Author

Reviewer #1: The suggested changes have been made correctly and completely. The commitment of the authors is remarkable. Great piece.

Reviewer #2: Dear authors,

I would like to thank you for this piece of work on this important.

I went through the text again and found out that you have addressed all the comments and answered questions that were raised from the previously submitted manuscript.

Reviewer #3: Excellent manuscript that offers a valuable contribution to the literature.

7. PLOS authors have the option to publish the peer review history of their article (what does this mean?). If published, this will include your full peer review and any attached files.

**Do you want your identity to be public for this peer review?** For information about this choice, including consent withdrawal, please see our Privacy Policy.

Reviewer #1: No

Reviewer #2: No

Reviewer #3: No
